# Low-Frequency Magnetoelectric Effects in Magnetostrictive– Piezoelectric Bilayers: Longitudinal and Bending Deformations

**Dmitry Filippov** [1,*] **, Ying Liu** [2,3] **, Peng Zhou** [3] **, Bingfeng Ge** [2,4] **, Jiahui Liu** [2,4] **, Jitao Zhang** [4] **, Tianjin Zhang** [3] **and Gopalan Srinivasan** [2,*]

1    Institute of Electronic and Information Systems, Yaroslav-the-Wise Novgorod State University, 173003 Veliky Novgorod, Russia
2    Physics Department, Oakland University, Rochester, MI 48309, USA; liuying.hube@outlook.com (Y.L.); bingfengGe@outlook.com (B.G.); Jiahui-Liu@outlook.com (J.L.)
3    Department of Materials Science and Engineering, Hubei University, Wuhan 430062, China; p_zhou@outlook.com (P.Z.); zhangtj@hubu.edu.cn (T.Z.)
4    College of Electrical and Information Engineering, Zhengzhou University of Light Industry, Zhengzhou 450002, China; zhang_jitao@outlook.com
*    Correspondence: dmitry.filippov@novsu.ru (D.F.); srinivas@oakland.edu (G.S.)

**Abstract:** A model for the low-frequency magnetoelectric (ME) effect that takes into consideration the bending deformation in a ferromagnetic and ferroelectric bilayer is presented. Past models, in general, ignored the influence of bending deformation. Based on the solution of the equations of the elastic theory and electrostatics, expressions for the ME voltage coefficients (MEVCs) and ME sensitivity coefficients (MESCs) in terms of the physical parameters of the materials and the geometric characteristic of the structure were obtained. Contributions from both bending and planar deformations were considered. The theory was applied to composites of PZT and Ni with negative magnetostriction, and Permendur, or Metglas, both with positive magnetostriction. Estimates of MEVCs and MESCs indicate that the contribution from bending deformation is significant but smaller than the contribution from planar deformations, leading to a reduction in the net ME coefficients in all the three bilayer systems.

**Keywords:** multiferroic composites; magnetostriction; piezoelectricity; magnetoelectric effect

## 1. Introduction

The nature of the coupling between magnetic and ferroelectric subsystems in composites of the two phases has been studied extensively during the past several years [1–3]. The interaction involves the transfer of strain produced by either a magnetic or electric field in one of the two phases to the other, which, in turn, leads to an electrical or a magnetic response, respectively. Composites consisting of a variety of ferromagnetic and ferroelectric phases were reported to show very strong magnetoelectric (ME) interactions when exposed to magnetic or electric fields at frequencies ranging from a few mHz to hundreds of GHz [1–6]. Studies involved bulk composites as well as thick-film- or thin-film-layered structures and nanocomposites in the form of nanopillars in a host matrix, core–shell particles, and core–shell nanofibers [7–10]. Layered magnetic–piezoelectric composites, in general, show a much stronger ME coupling, compared with bulk composites [11]. One of the important advantages of layered structures is the ease of their fabrication process, and it is possible to use ferromagnetic metals or alloys with high magnetostriction, such as Permendur, Terfenol-D, Metglas, etc., whereas in bulk composites, the choice for the ferromagnetic phase is restricted to high resistivity oxides such as nickel ferrite or cobalt ferrite with relatively low magnetostriction [11]. Ferrites are poor insulators; therefore, their use in bulk composites leads to large leakage currents, which lead to the weakening of ME interactions [12,13]. In the layered structures, however, the ferromagnetic layers are well insulated with piezoelectric layers, and as a result, the leakage currents are negligibly small.

A widely used technique in characterizing the nature of ME interactions is the measurements of low-frequency ME voltage coefficient (MEVC) that involves applying an AC magnetic field ($h_{ac}$) and measuring the voltage ($V_{ac}$) produced across the ferroelectric layer of thickness $t^p$, and MEVC = $V_{ac}/(t^p h_{ac})$ is a measure of the strength of ME coupling. Several models were developed in the past for the phenomenon in bilayer structures [14–20]. These include low-frequency ME effects due to longitudinal vibrations caused by the AC magnetic field [14–16], the influence of texture and residual stress in the layers [17,18], and the size of the layers and the corresponding demagnetization factor on ME coupling [19]. Related modeling efforts of interest include direct and converse ME effects in laminates [21], symmetric trilayer composites [22], and nanocomposites [23]. In addition to the longitudinal deformation caused by a magnetic field, bending deformation is also present in the composite. The ME effect at the resonance modes of bending deformations was modeled and studied in several bilayers [24–26]. Here, we discuss the first model for low-frequency ME effects that takes into account the bending deformation in a ferromagnetic–ferroelectric bilayer. Past theories for the low-frequency did not consider the influence of bending deformation on MEVC [14–20,27–30]. A refined model that considers bending deformation is also of importance due to interests in the utility of the phenomenon for applications such as pico-Tesla magnetic sensors and energy harvesters [2,31]

The theory discussed here is based on the equations of elasticity and electrostatics. Expressions were obtained for the MEVC in terms of the physical parameters of materials and the geometric characteristics of the structure. Contributions due to the longitudinal and bending deformations and their dependence on the geometric parameters of the structure were analyzed. The theory was applied to three representative bilayers: Nickel–PZT, Permendur–PZT, and Metglas–PZT. Nickel has negative magnetostriction in the direction of the longitudinal field, whereas both Permendur, a ferromagnetic alloy, and Metglas have positive magnetostriction. The MEVC was estimated as a function of the thickness of the ferromagnetic and ferroelectric layers. It is shown that contribution to ME voltage due to bending can be as high as 50% of the longitudinal deformation and has a 180 deg phase difference and always results in a decrease in the net MEVC. We also show the predicted variation in the ME sensitivity coefficient (MESC), defined as the ratio of $h_{ac}$ to $V_{ac}$, as a function of layer thickness for the three bilayers.

## 2. Model and Method of the Calculations

For the model, we considered a bilayer structure, as shown in Figure 1. The origin of the coordinate system is compatible with the center of the sample, and the *X*-axis (1) is compatible with the interface between the piezoelectric layer and the magnetic layer. We assumed that the sample's thickness is much smaller than its length and width. The elastostatic and electrostatic equations for the piezoelectric and magnetostrictive phases in this approximation have the following form:

$$S_1^p = \frac{1}{Y^p} T_1^p + d_{31}^p E_3, \tag{1}$$

$$S_1^m = \frac{1}{Y^m} T_1^m + q_{11}^m H_1 \tag{2}$$

$$D_3^p = \varepsilon_{33}^p E_3 + d_{31}^p T_1^p, \tag{3}$$

where $S_1^p$ and $S_1^m$ are strain tensor components of piezoelectric and magnetostrictive layers; $Y^p$ and $Y^m$ are their Young's moduli; $E_3$ and $D_3^p$ are components of the vector of the electric field and electric induction; $T_1^p$ and $T_1^m$ are the stress tensor components of the piezoelectric and magnetostrictive phases; $d_{31}^p$ and $q_{11}^m$ are piezoelectric and piezomagnetic coefficients; and $\varepsilon_{33}^p$ is the component of the permittivity.

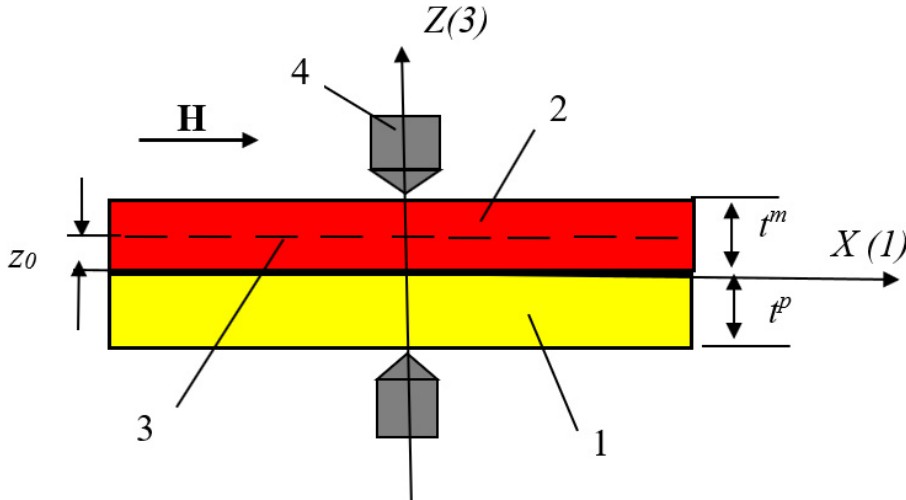

**Figure 1.** Schematic drawing of a magnetostrictive–piezoelectric bilayer showing the piezoelectric layer (1), magnetostrictive layer (2), neutral plane (3), and electrodes (4).

When the sample is placed in a magnetic field, tensile deformations occur in the case of positive magnetostriction (Permendur, D-Terfenol, or Metglas), or compression if the magnetic layer has negative magnetostriction (nickel, nickel ferrite). By means of mechanical coupling through the interface, these deformations are transferred to the piezoelectric phase, because of which the sample can experience longitudinal deformations such as tension or compression. Since these deformations are not axial, they also lead to a bending moment and bending deformations. Since the layers are assumed to be thin, we can assume that longitudinal strains are uniform throughout the layer volume, i.e., the following equality holds:

$$S_1^m = S_1^p = S_1, \tag{4}$$

The equilibrium condition of the sample–namely, the equality to zero the *X* projection of the force, yields the following equation:

$$T_1^m t^m + T_1^p t^p = 0. \tag{5}$$

Expressing the components of the stress tensor from Equations (1) and (2) and substituting the obtained expressions into Equation (5), we obtain the following expression:

$$(Y^m t^m + Y^p t^p)S_1 - Y^m t^m q_{11}^m H_1 - Y^p t^p d_{31}^p E_3 = 0. \tag{6}$$

Hence, for longitudinal deformations, we obtain an expression in the following form:

$$S_1 = \frac{Y^m t^m q_{11}^m H_1 + Y^p t^p d_{31}^p E_3}{\overline{Y}t}. \tag{7}$$

where $\overline{Y} = \frac{Y^m t^m + Y^p t^p}{t}$ is the average value of Young's modulus of the structure, and $t = t^m + t^p$ is the total thickness of the bilayer.

Substituting the obtained expression into Equation (3) and using the open-circuit condition, which in this case has the form $D_3^p = 0$, we obtain for the electric field induced in the piezoelectric due to longitudinal deformations the following expression:

$$E_{3,long} = -\frac{Y^p d_{31}^p q_{11}^m}{\varepsilon_{33}^p \left(1 - k_p^2 \left(1 - \frac{Y^p t^p}{Yt}\right)\right)} \frac{Y^m t^m}{Yt} H_1, \tag{8}$$

Using the definition of the MEVC in the form $\alpha_E = \frac{E_{3,plan}}{H_1}$, we obtain the following expression for the contribution to it from longitudinal deformations:

$$\alpha_{E,long} = -\frac{Y^p d_{31}^p q_{11}^m}{\varepsilon_{33}^p \left(1 - k_p^2 \left(1 - \frac{Y^p t^p}{\overline{Y}t}\right)\right)} \frac{Y^m t^m}{\overline{Y}t}, \tag{9}$$

In Equation (9), for the MEVC, the parameter $k_p^2 \ll 1$; therefore, this expression can be simplified by writing it in the following form:

$$\alpha_{E,long} = -\frac{Y^p d_{31}^p q_{11}^m}{\varepsilon_{33}^p} \frac{Y^m t^m}{\overline{Y}t}, \tag{10}$$

Along with the MEVC, which is the main ME parameter characterizing the linear ME effect, we can use one more parameter to characterize the magnetic-field-to-electric-field conversion efficiency. This parameter—namely, the ME sensitivity coefficient, is equal to the ratio of the magnitude of the induced electric voltage $U_{plan} = E_{3,plan} t^p$ to the magnitude of the alternating magnetic field, i.e., $\beta_{U,long} = \frac{U_{long}}{H_1}$. Using Equation (8), we obtain the following expression for the ME sensitivity coefficient (MESC):

$$\beta_{U,long} = -\frac{Y^p d_{31}^p q_{11}^m}{\varepsilon_{33}^p \left(1 - k_p^2 \left(1 - \frac{Y^p t^p}{\overline{Y}t}\right)\right)} \frac{Y^m t^m t^p}{\overline{Y}t}, \tag{11}$$

or in a simplified form

$$\beta_{U,long} = -\frac{Y^p d_{31}^p q_{11}^m}{\varepsilon_{33}^p} \frac{Y^m t^m t^p}{\overline{Y}t}. \tag{12}$$

Equations (9)–(12) make it possible to analyze the dependence of the MEVC and MESC due to longitudinal deformation on the physical parameters of the magnetostrictive and piezoelectric phases and their layer thicknesses.

When considering the bending deformations, we used the Bernoulli hypothesis [32]. We assumed that the bonding between the layers is ideal and, consequently, for the deformations of the piezoelectric and magnetic layers, the following relation holds:

$$S_1 = \frac{(z - z_0)}{\rho}, \tag{13}$$

where $z_0$ is a coordinate of the neutral line, and $\rho$ is the radius of curvature of the neutral line, which is related to the bending moment by the following relation:

$$\frac{1}{\rho} = \frac{M_y}{Y^m J_{z0}^m + Y^p J_{z0}^p}, \tag{14}$$

The following notations were introduced here: $M_y = \int_0^W dy \cdot (\int_{-t^p}^0 (z - z_0) T_1^p dz + \int_0^{t^m} (z - z_0) T_1^m dz)$ is a bending moment; $J_{z0}^m$ and $J_{z0}^p$ are inertia moments of sections about the neutral axis $z_0$. These inertia moments, according to Steiner's theorem, are determined by the following expressions:

$$J_{z0}^m = \frac{1}{12} W(t^m)^3 + W t^m (t^m/2 - z_0)^2, \tag{15}$$

$$J_{z0}^p = \frac{1}{12} W(t^p)^3 + W t^p (t^p/2 + z_0)^2. \tag{16}$$

The position of the neutral line is determined from the condition that the X-projection of the force is equal to zero. For our case, this condition has the following form:

$$\int_{-t^p}^{0} T_1^p dz + \int_{0}^{t^m} T_1^m dz = 0 \tag{17}$$

Substituting into Equation (15), the expression for the components of the stress tensor, which can be obtained from Equations (1) and (2), and assuming the external influences to be weak, for the neutral line coordinate $z_0$, we obtain the following expression:

$$z_0 = \frac{1}{2} \frac{Y^m (t^m)^2 - Y^p (t^p)^2}{\overline{Y} t}, \tag{18}$$

The neutral line in the bilayer structure can lie in the piezoelectric layer or in the magnetostrictive layer. If the neutral line is in a piezoelectric layer, then, in this case, one part of the piezoelectric that lies above the neutral line undergoes tension (compression), while the other part undergoes compression (tension). As a result, the resulting electric fields in different parts of the piezoelectric have opposite directions, because of which the total electric field decreases. If the neutral layer is in a magnetostrictive layer, then the bending moments arising under the action of the magnetic field in the parts located on opposite sides of the neutral line have opposite directions, because of which the total bending moment decreases. The maximum ME response is in the case when the neutral line is located at the interface between the magnetostrictive layer and piezoelectric layer, i.e., when the neutral line coordinate is equaled $z_0 = 0$. According to Equation (18), this occurs when the following relation between the thicknesses of the magnetostrictive and the piezoelectric layers applies:

$$Y^m (t^m)^2 = Y^p (t^p)^2. \tag{19}$$

The induced electric field in the piezoelectric layer because of bending deformations can be found, similar to the case of the longitudinal deformation, from the open-circuit condition, which, in our case, has the form $D_3^p = 0$. Using this condition and Equations (1) and (3) we obtain

$$E_{3,bend}(z) = -\frac{1}{\varepsilon_{33}^p} d_{31}^p \left( Y^p S_1^p - Y^p d_{31}^p E_3(z) \right). \tag{20}$$

or after simple transformations, using Equations (13) and (14), we obtain

$$E_{3,bend}(z) = -\frac{d_{31}^p Y^p (z - z_0)}{\varepsilon_{33}^p D \left( 1 - k_p^2 \right)} (t^m / 2 - z_0) Y^m t^m q_{11}^m H_1. \tag{21}$$

where $D = \frac{Y^m J_{z0}^m + Y^p J_{z0}^p}{W}$ is cylindrical bending stiffness.

For electric voltage between electrodes of the sample due to bending deformations, we obtain the following equation:

$$U_{bend} = \int_{-t^p}^{0} E_{3,bend}(z) dz \tag{22}$$

Substituting Equation (21) into Equation (22) and integrating, we obtain

$$U_{bend} = \frac{d_{31}^p Y^p t^p}{\varepsilon_{33}^p D \left( 1 - k_p^2 \right)} (0.5 t^p + z_0)(0.5 t^m - z_0) Y^m t^m q_{11}^m H_1 \tag{23}$$

One may obtain expressions for the MEVC and the MESC using its definition $\alpha_{E,bend} = \frac{\langle E_{3,bend} \rangle}{H_1}$ and $\beta_{U,bend} = \frac{U_{bend}}{H_1}$, where $\langle E_{3,bend} \rangle = \frac{U_{bend}}{t^p}$ is the average value electric field induced by bending deformations. Using these definitions and Equation (23), we obtain the following expressions for MEVC and MESC:

$$\alpha_{E,bend} = \frac{d_{31}^p q_{11}^m Y^p Y^m t^m}{\varepsilon_{33}^p D \left(1 - k_p^2\right)} (0.5t^p + z_0)(0.5t^m - z_0) \tag{24}$$

$$\beta_{U,bend} = \frac{d_{31}^p q_{11}^m Y^p Y^m t^m}{\varepsilon_{33}^p D \left(1 - k_p^2\right)} t^p (0.5t^p + z_0)(0.5t^m - z_0) \tag{25}$$

These equations can be rewritten in simplified forms, using the fact that the square of the electromechanical coupling parameter $k_p^2 \ll 1$. It yields the following expressions:

$$\alpha_{E,bend} = \frac{d_{31}^p q_{11}^m Y^p Y^m t^m}{\varepsilon_{33}^p D} (0.5t^p + z_0)(0.5t^m - z_0), \tag{26}$$

$$\beta_{U,bend} = \frac{d_{31}^p q_{11}^m Y^p Y^m t^m}{\varepsilon_{33}^p D \left(1 - k_p^2\right)} t^p (0.5t^p + z_0)(0.5t^m - z_0), \tag{27}$$

Equations (24)–(27) can be used to estimate the dependence of the MEVC and the MESC on the physical and geometrical parameters of the bilayer structure.

## 3. Results and Discussions

The expressions in the previous section for contributions to MEVC and MESC from longitudinal and bending deformation in a ferromagnetic and ferroelectric bilayer facilitate the estimation of the net ME coefficients. The net MEVC $\alpha_{E,net}$ and MESC $\beta_{U,net}$ are given by

$$\alpha_{E,net} = \alpha_{E,long} + \alpha_{E,bend}, \tag{28}$$

$$\beta_{U,net} = \beta_{U,long} + \beta_{U,bend}, \tag{29}$$

It should be noted that the contributions from longitudinal and bending deformations enter the sums with opposite signs. In the case of longitudinal oscillations, deformations arising in the magnetostrictive layer under the action of a magnetic field cause deformations of the same sign in the piezoelectric layer. For example, in a magnetic layer with positive magnetostriction, the tensile strain that occurs in the layer when transmitted through the interface causes tensile deformations in the piezoelectric. In the case of bending, however, a tensile deformation in the magnetic layer causes compression deformation in the piezoelectric layer, resulting in an electric field directed opposite to the electric field caused by longitudinal deformations. Both contributions are proportional to the product of the piezoelectric coefficient $d_{31}^p$, the piezomagnetic coefficient $q_{11}^m$ and Young's modulus of the piezoelectric $Y^p$, and are inversely proportional to the permittivity $\varepsilon_{33}^p$. The contributions do not depend on the width and length of the sample but depend on the thickness of the piezoelectric and magnetic layers.

Next, we applied the theory to three representative bilayer composites with Ni, Permendur (an alloy of Fe, Co, and V), or Metglas for the ferromagnetic layer and PZT for the ferroelectric layer. Nickel has negative longitudinal magnetostriction, whereas this is positive for Permendur and Metglas. The piezomagnetic coefficients for the ferromagnets and the piezoelectric coefficient for PZT are listed in Table 1 [33].

**Table 1.** Parameters of materials of composite structures [33].

| Material | Young's Modulus Y, GPa | Piezomodules $d_{31}$, pC/N; and $q_{11}$, ppm/Oe | Permittivity $\varepsilon$ |
|---|---|---|---|
| PZT | 66.7 | $d_{31} = -175$ | 1750 |
| Ni | 215 | $q_{11} = -0.06$ | - |
| Pe | 207 | 0.1 | - |
| Metglas | 110 | 0.3 | |

Figures 2 and 3 show the MEVC and MESC dependencies, respectively, for the nickel–PZT bilayer. The ME voltage coefficient is shown as a function of Ni thickness in Figure 2 for a fixed PZT thickness of 0.5 mm. As can be seen from Equation (10) and Figure 2, the MEVC caused by the longitudinal deformations increase with the increase in the thickness of the Ni layer and attains the limiting value for $t^m \gg t^p$.

$$\left(\alpha_{E,long}\right)_{t^m \to \infty} = -\frac{Y^p d_{31}^p q_{11}^m}{\varepsilon_{33}^p}. \tag{30}$$

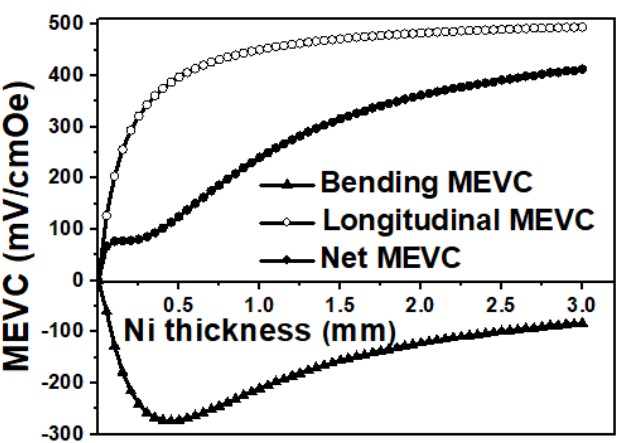

**Figure 2.** Estimated dependence of MEVC in a Ni-PZT bilayer on the magnetostrictive layer thickness. The piezoelectric layer thickness $t^p = 0.5$ mm.

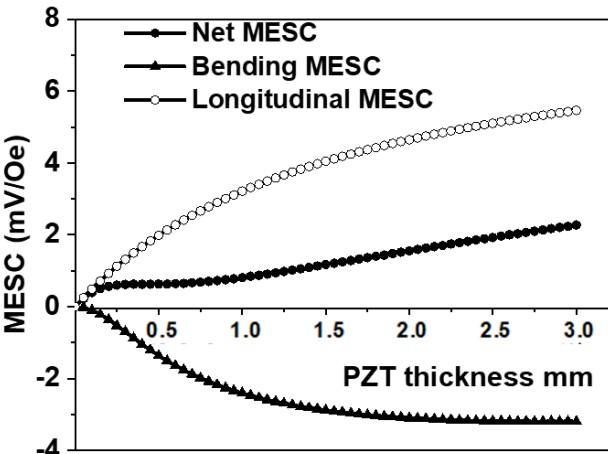

**Figure 3.** Dependence of MESC in a Ni-PZT bilayer on the piezoelectric layer thickness. The magnetostrictive layer thickness $t^m = 0.5$ mm.

The MEVC associated with bending deformations is zero at $t^m = 0$, then increases until it reaches a maximum when the neutral line coincides with the ferromagnetic–piezoelectric

interface and then decreases. As already noted, these contributions have different signs, and the contribution from longitudinal deformations exceeds the contribution from bending deformations over the entire range of Ni thickness. The total MEVC increases with an increase in the thickness of the Ni layer. Thus, the overall effect of the bending deformation is a reduction in total MEVC. This reduction is approximately 75% of the contribution from the longitudinal deformations for Ni thickness of 0.5 mm, and it decreases to 20% for $t^m = 3$ mm.

Figure 3 shows the predicted variation in MESC with the thickness of the PZT layer. The Ni thickness is assumed to be 0.5 mm. The MESC equals zero at $t^p = 0$. The MESC caused by longitudinal deformations increases with the increase in PZT layer thickness and is predicted to attain saturation at $t^p \gg t^m$. This saturation value of MESC equals

$$\left( \beta_{U,long} \right)_{t^p \to \infty} = -\frac{Y^m d_{31}^p q_{11}^m}{\varepsilon_{33}^p} t^m. \tag{31}$$

The MESC due to bending deformations increases with the increase in $t^p$, then it reaches a maximum value and then slowly decreases with a further increase in the thickness of the piezoelectric layer. The net MESC increases at first with increases in the thickness of the piezoelectric layer, and then there is a small plateau in its value. The presence of a plateau is due to the fact that the rate of increase in the bending MESC with the thickness of the ferroelectric layer and of the rate at which the longitudinal MESC increases have the opposite signs, and as a result, the net MESC remains unchanged. With a further increase in PZT thickness, the bending MESC begins to decrease, and the longitudinal MESC continues to increase; as a result, the Net MESC increases again and tends to saturation at $t^p \gg t^m$. This saturation value of the total MESC is given by Equation (31).

Similar estimates of MEVC and MESC for a bilayer of Permendur and PZT are shown in Figures 4 and 5, respectively. The overall features in the results are similar to the case of Ni–PZT. There is a sign reversal in the contributions from longitudinal and bending deformations to MEVC in Figure 4, which is due to the positive magnetostriction and piezomagnetic coefficient for Permendur. A peak in the ME voltage due to bending is seen for $t^m = 0.5$ mm, which is 63% of the contribution from longitudinal deformations. The contribution from the longitudinal deformation dominates for higher $t^m$ values and the saturation value for MEVC is much higher than for Ni–PZT due to the higher $q_{11}$ value for Permendur. Figure 5 shows MESC vs. $t^p$ for the bilayer for $t^m = 0.5$ mm. The sign reversal in the contributions from longitudinal and bending deformations seen for MEVC also occurs in the results for MESC. For $t^p = 3$ mm, the MESC value is a factor of two higher than for the case of Ni–PZT and is due to the high $q_{11}$ value for Permendur.

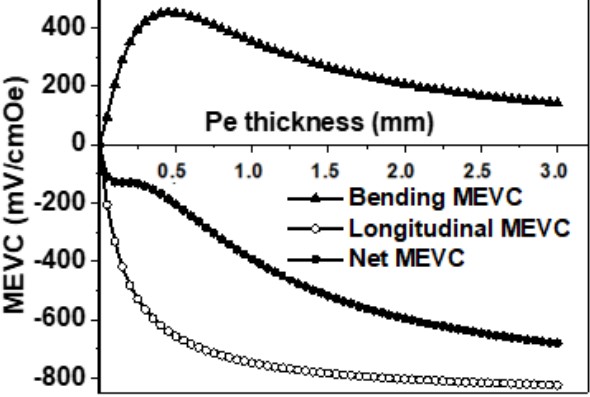

**Figure 4.** Results on estimated MEVC vs. $t^m$, as in Figure 2 for a bilayer of Permendur (Pe) and PZT for $t^p = 0.5$ mm.

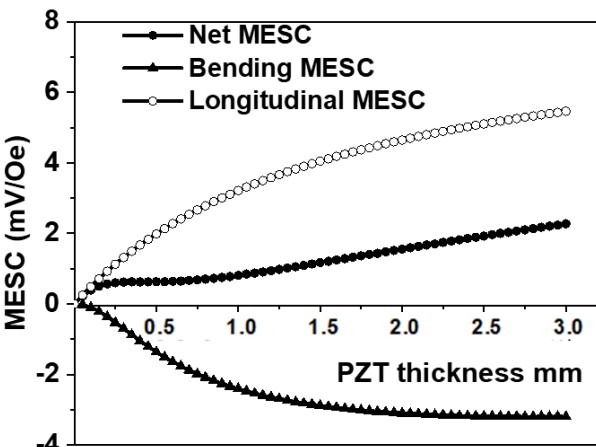

**Figure 5.** Estimated MESC vs. $t^p$ for a bilayer of Pe-PZT. The magnetostrictive layer thickness $t^m = 0.5$ mm.

Theoretical estimates on MEVC and MESC for a bilayer of Metglas and PZT are shown in Figures 6 and 7, respectively. Since Metglas has very high permeability, essential for the confinement of magnetic fields, and high piezomagnetic coefficient, the composite with PZT is of interest for applications for high-sensitivity magnetic sensors and in energy harvesting. In Metglas–PZT contributions from bending and longitudinal deformations to MEVC and MESC are similar to the case of Permendur–PZT. The theory predicts a value of 1.6 V/cm Oe for MEVC for $t^m = 3$ mm, compared with 0.4 V/cm Oe and 0.7 V/cm Oe for Ni–PZT and Pe–PZT, respectively. This is due to the fact that the MEVC is directly proportional to the piezomagnetic coefficient, listed in Table 1. The $q_{11}$ value for Ni is the lowest amongst the three bilayer systems considered here, and it is the highest for Metglas. Thus, the MEVC for Metglas–PZT is expected to be the highest, followed by Permendur–PZT and Ni–PZT. A similar enhancement in MESC is seen for Metglas–PZT in the results of MESC vs. $t^p$ in Figure 7.

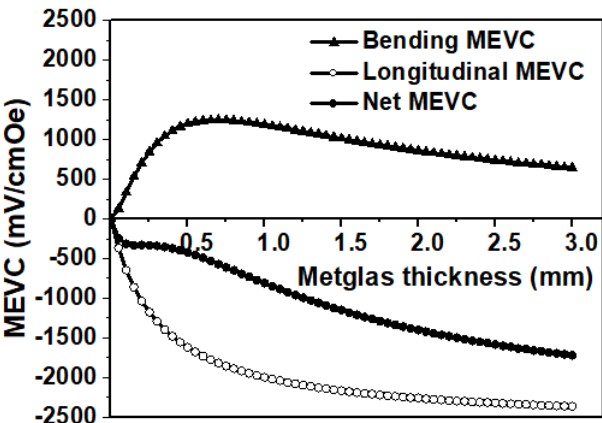

**Figure 6.** Results on estimated MEVC vs. $t^m$, as in Figure 2 for a bilayer of Metglas and PZT for $t^p = 0.5$ mm.

The theory developed in Section 3 and its application to specific bilayers in Section 4 clearly indicate the need to consider the contribution of bending deformation to the low-frequency ME response of the composites to a magnetic field. Bending deformation is shown to weaken the strength of ME coupling and the reduction in the net MEVC and MESC depends on the piezoelectric and piezomagnetic coefficients and the thickness of the ferromagnetic and ferroelectric layers. The bending-related reduction in MEVC tends to be smaller than the contribution from the longitudinal deformation only when the thickness of the piezoelectric layer $t^p$ is much higher than $t^m$. Although bending deformation-related

reduction in MEVC is always present in a bilayer composite, it can be completely eliminated in a symmetric trilayer composite. The theory developed is of importance for applications such as highly sensitive magnetic sensors and for energy harvesting [1–4,34].

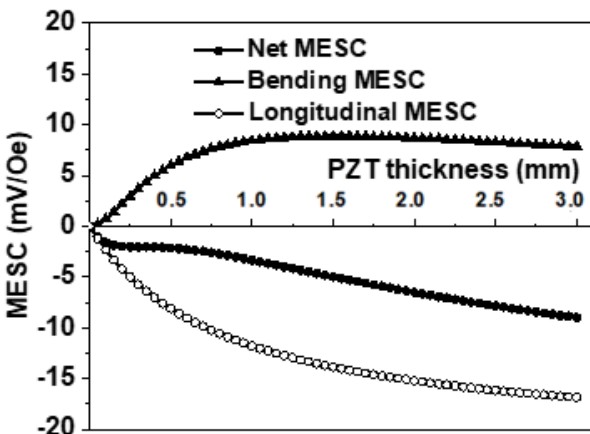

**Figure 7.** Estimated MESC vs. $t^p$ for a bilayer of Metglas–PZT. The magnetostrictive layer thickness $t^m = 0.5$ mm.

## 4. Conclusions

In bilayer magnetostrictive–piezoelectric structures, the magnetoelectric effect is associated with two types of deformations. These are longitudinal deformations and bending deformations and occur when the bilayer is subjected to a magnetic field. The contributions to the total MEVC from longitudinal deformations and bending deformations have opposite signs. The total MEVC is zero when the thickness of the magnetostrictive layer equals zero, and it increases with an increase in the thickness of the magnetic layer. The dependence of the total MEVC will have a small plateau in the range, where the value of MEVC from bending deformations has a maximum and then increases again with increasing $t^m$. The ME sensitivity coefficient, a parameter of importance for device applications, equals zero at the zero value of piezoelectric layer thickness. With increasing $t^p$, the total MESC increases at first, and then there is a plateau where the value of MESC from bending deformations has a maximum. Then, it increases again and tends to saturation at $t^p \gg t^m$.

The primary objective of this work was to address the shortcomings in past theories for low-frequency magnetoelectric (ME) effects in a bilayer of a ferromagnet and a ferroelectric. Another objective of the work was to provide a roadmap for experimentalists to utilize the results of our model to estimate the expected ME coefficients for known parameters for the ferroic phases including the piezoelectric and piezomagnetic coefficients and their thicknesses. This aspect was demonstrated by applying the theory to three representative bilayer systems. The ultimate goal is for experimentalists to choose appropriate ferroic systems to achieve the desired low-frequency ME response and compare the measured ME coefficients with the results of our model.

**Author Contributions:** All the authors contributed to this work. Data curation, D.F., G.S., T.Z. and J.Z.; formal analysis, Y.L., P.Z., B.G. and J.L.; funding acquisition, G.S., T.Z. and J.Z.; methodology, D.F.; project administration, G.S. and T.Z.; writing—original draft preparation, D.F. and G.S. All authors have read and agreed to the published version of the manuscript.

**Funding:** This research was funded at Oakland University by grants from the US Air Force Office of Scientific Research (AFOSR) Award No. FA9550-20-1-0114 and the US National Science Foundation (DMR-1808892, ECCS-1923732). Ying Liu was supported by a fellowship from the Chinese Scholarship Council. The research at Hubei University was supported by the China Postdoctoral Science Foundation (No. 2020M672315) and the Program of Hubei Key Laboratory of Ferro- and Piezoelectric Materials and Devices (No. K202013). The research at the Zhengzhou University of Light Industry

was supported by a grant from the National Natural Science Foundation of China (NSFC, Grant No. 61973279).

**Data Availability Statement:** Data are available from the corresponding author upon reasonable request.

**Conflicts of Interest:** The authors declare no conflict of interest.

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
