# Peer review of "Low-Frequency Magnetoelectric Effects in Magnetostrictive–Piezoelectric Bilayers: Longitudinal and Bending Deformations"

_jcs, doi:10.3390/jcs5110287_

Round 1
Reviewer 1 Report
The authors presented a model for low-frequency magnetoelectric (ME) effect that takes into consideration the bending deformation in a ferromagnetic and ferroelectric bilayer. It is theoretically examined three kinds of PZT composites in detail.
1. As for these theoretical results, is it had the comparison with the laboratory (experimental) finding?
2. Explain the cause of the difference of MEVC between Metglas-PZT, Ni-PZT, and Pe-PZT, which was described in L252-254.
3. L229-230 : "The total MESC increases at first with increases the thickness of the piezoelectric layer, then there is a small plateau in its value. " Explain the physical origin of this plateau.
Author Response
Report 1
We are grateful to the reviewer for the comments. The manuscript has been revised taking into consideration all of the comments. Our response to specific comments are given below.
The authors presented a model for low-frequency magnetoelectric (ME) effect that takes into consideration the bending deformation in a ferromagnetic and ferroelectric bilayer. It is theoretically examined three kinds of PZT composites in detail.
- As for these theoretical results, is it had the comparison with the laboratory (experimental) finding?
Response: The following is added to the conclusion part. “The primary objective of this work was to address the shortcomings in past theories for low frequency magneto-electric (ME) effects in a bilayer of a ferromagnet and a ferroelectric. Another objective of the work was to provide a roadmap for experimentalists to utilize the results of our model to estimate the expected ME coefficients for known parameters for the ferroic phases including the piezoelectric and piezomagnetic coefficients and their thicknesses. This aspect was demonstrated by applying the theory to three representative bilayer systems. The ultimate goal is for experimentalists to choose appropriate ferroic systems to achieve the desired low frequency ME response and compare the measured ME coefficients with the results of our model.”
- Explain the cause of the difference of MEVC between Metglas-PZT, Ni-PZT, and Pe-PZT, which was described in L252-254.
Response: The theory predicts a value of 1.6 V/cm Oe for MEVC for tm = 3 mm compared to 0.4 V/cm Oe and 0.7 V/cm Oe for Ni-PZT and Pe-PZT, respectively. This is due to the fact that the MEVC is directly proportional to the piezomagnetic coefficient listed in Table 1. The q11 value for Ni is the lowest amongst the three bilayer systems considered here and it is the highest for Metglas. Thus the MEVC for Metglas-PZT is expected to be the highest, followed by Permendur-PZT, and Ni-PZT.
- L229-230 : "The total MESC increases at first with increases the thickness of the piezoelectric layer, then there is a small plateau in its value. " Explain the physical origin of this plateau.
Response: The presence of a plateau is due to the fact that the rate of increase of the bending MESC with the thickness of the ferroelectric layer and of the rate at which the longitudinal MESC increases have the opposite signs and as a result the net MESC remains unchanged. With a further increase of PZT thickness, the bending MESC begins to decrease and the longitudinal MESC continues to increase and as a result, the Net MESC is increasing again and tends to saturation at tp >> tm.
Reviewer 2 Report
This is an interesting paper. It reports low-frequency magnetoelectric effects in magnetostrictive-piezoelectric bilayers: longitudinal and bending deformations. In my opinion, the subject of the manuscript concerns very compelling issues. The topics discussed in this manuscript are fully match the journal. The authors described a model for the low-frequency magnetoelectric (ME) effect that takes into consideration the bending deformation in a ferromagnetic and ferroelectric bilayer. Literature data show that past models in general ignored the influence of the bending deformation, which, in my opinion, can significantly affect the obtained results. In this manuscript, the authors, based on the solution of the equations of the elastic theory and electrostatics, obtain expressions for the ME voltage coefficients (MEVC) and ME sensitivity coefficients (MESC) in terms of the physical parameters of the materials and the geometric characteristic of the structure. The contributions from both bending and planar deformations are considered. According to the authors of the presented research, the theory is applied to composites of PZT and Ni with negative magnetostriction, Permendur, or Metgals with positive magnetostriction.
Personally, I think the title of the manuscript is acceptable. The title describes fully the subject matter of the article. In my judgment, the information presented in the manuscript is useful and reliable.
Personally, I think that the figures and tables included in the manuscript are of good quality and do not require no corrections.
From my point of view, there are adequate and appropriate and adequate references to previous and related and previous work in this manuscript, but the references are not fully prepared in accordance with the journal’s requirements. References should be corrected as required by the journal.
I am convinced that the paper can be published.
Author Response
Response to Report 2
This is an interesting paper. It reports low-frequency magnetoelectric effects in magnetostrictive-piezoelectric bilayers: longitudinal and bending deformations. In my opinion, the subject of the manuscript concerns very compelling issues. The topics discussed in this manuscript are fully match the journal. The authors described a model for the low-frequency magnetoelectric (ME) effect that takes into consideration the bending deformation in a ferromagnetic and ferroelectric bilayer. Literature data show that past models in general ignored the influence of the bending deformation, which, in my opinion, can significantly affect the obtained results. In this manuscript, the authors, based on the solution of the equations of the elastic theory and electrostatics, obtain expressions for the ME voltage coefficients (MEVC) and ME sensitivity coefficients (MESC) in terms of the physical parameters of the materials and the geometric characteristic of the structure. The contributions from both bending and planar deformations are considered. According to the authors of the presented research, the theory is applied to composites of PZT and Ni with negative magnetostriction, Permendur, or Metgals with positive magnetostriction.
Personally, I think the title of the manuscript is acceptable. The title describes fully the subject matter of the article. In my judgment, the information presented in the manuscript is useful and reliable.
Personally, I think that the figures and tables included in the manuscript are of good quality and do not require no corrections.
From my point of view, there are adequate and appropriate and adequate references to previous and related and previous work in this manuscript, but the references are not fully prepared in accordance with the journal’s requirements. References should be corrected as required by the journal.
I am convinced that the paper can be published.
Response: We are grateful to the reviewer for the above comments. We have provided references in the format required by the journal.
Reviewer 3 Report
Revision of “Low-Frequency Magnetoelectric Effects in Magnetostrictive- Piezoelectric Bilayers: Longitudinal and Bending Deformations”
The manuscript under review devoted to a model for low-frequency magnetoelectric (ME) effect that takes into consideration the bending deformation in a ferromagnetic and ferroelectric bilayer. Providing of such investigations is very important both from an academic point of view (giving new knowledge about the nature of the objects under study) and economic (reducing the costs of industrial companies when obtaining new materials for various areas of the civil sector).
Using the proposed model, the authors obtained the following conclusions. In bilayer magnetostrictive-piezoelectric structures, the magnetoelectric effect is associated with two types of deformation. These are longitudinal deformations and bending deformations and occur when the bilayer is subjected to a magnetic field. The contributions to the total MEVC from longitudinal deformations and bending deformations have opposite signs. The total MEVC is zero when the thickness of the magnetostrictive layer equals zero and it increases with increase in the thickness of the magnetc layer. The dependence of the total MEVC will have a small plateau in the range where the value of MEVC from bending deformations has a maximum and then increases again with increasing tm. The ME sensitivity coefficient, a parameter of importance for device applications, equals zero at the zero value of piezoelectric layer thickness. With increasing tp, the total MESC increases at first, then there is a plateau where the value of MESC from bending deformations has a maximum. Then it increases again and tends to saturation at tp>>tm.
In manuscript all necessary information is captured by 7 figures. There are 34 references, all of them are adequate and are reflected in the text.
After getting acquainted with the presented manuscript, a few small questions remained:
- Formula 4 apparently contains an error when using superscripts.
- In Section 3, the numbering of the given formulas is violated.
The obtained results are important both for understanding the physical processes that occur in real objects and for the development of new materials. The described manuscript is sufficient, comprehensive and it corresponds to the field of the Journal «Composites Science». It may be accepted after minor revision.
Author Response
Report 3
We are grateful to the referee for the comments. Our response tp specific comments are given below.
The manuscript under review devoted to a model for low-frequency magnetoelectric (ME) effect that takes into consideration the bending deformation in a ferromagnetic and ferroelectric bilayer. Providing of such investigations is very important both from an academic point of view (giving new knowledge about the nature of the objects under study) and economic (reducing the costs of industrial companies when obtaining new materials for various areas of the civil sector).
Using the proposed model, the authors obtained the following conclusions. In bilayer magnetostrictive-piezoelectric structures, the magnetoelectric effect is associated with two types of deformation. These are longitudinal deformations and bending deformations and occur when the bilayer is subjected to a magnetic field. The contributions to the total MEVC from longitudinal deformations and bending deformations have opposite signs. The total MEVC is zero when the thickness of the magnetostrictive layer equals zero and it increases with increase in the thickness of the magnetc layer. The dependence of the total MEVC will have a small plateau in the range where the value of MEVC from bending deformations has a maximum and then increases again with increasing tm. The ME sensitivity coefficient, a parameter of importance for device applications, equals zero at the zero value of piezoelectric layer thickness. With increasing tp, the total MESC increases at first, then there is a plateau where the value of MESC from bending deformations has a maximum. Then it increases again and tends to saturation at tp>>tm.
In manuscript all necessary information is captured by 7 figures. There are 34 references, all of them are adequate and are reflected in the text.
After getting acquainted with the presented manuscript, a few small questions remained:
- Formula 4 apparently contains an error when using superscripts.
- In Section 3, the numbering of the given formulas is violated.
The obtained results are important both for understanding the physical processes that occur in real objects and for the development of new materials. The described manuscript is sufficient, comprehensive and it corresponds to the field of the Journal «Composites Science». It may be accepted after minor revision.
Response: We corrected Equation 4 and the numbering of the equations in Section 3.
Round 2
Reviewer 1 Report
The revised article deserves publication.
Reviewer 2 Report
I approve the article for publication in the journal.
Reviewer 3 Report
Revision of “Low-Frequency Magnetoelectric Effects in Magnetostrictive- Piezoelectric Bilayers: Longitudinal and Bending Deformations”
The manuscript under review devoted to a model for low-frequency magnetoelectric (ME) effect that takes into consideration the bending deformation in a ferromagnetic and ferroelectric bilayer. Providing of such investigations is very important both from an academic point of view (giving new knowledge about the nature of the objects under study) and economic (reducing the costs of industrial companies when obtaining new materials for various areas of the civil sector).
Using the proposed model, the authors obtained the following conclusions. In bilayer magnetostrictive-piezoelectric structures, the magnetoelectric effect is associated with two types of deformation. These are longitudinal deformations and bending deformations and occur when the bilayer is subjected to a magnetic field. The contributions to the total MEVC from longitudinal deformations and bending deformations have opposite signs. The total MEVC is zero when the thickness of the magnetostrictive layer equals zero and it increases with increase in the thickness of the magnetc layer. The dependence of the total MEVC will have a small plateau in the range where the value of MEVC from bending deformations has a maximum and then increases again with increasing tm. The ME sensitivity coefficient, a parameter of importance for device applications, equals zero at the zero value of piezoelectric layer thickness. With increasing tp, the total MESC increases at first, then there is a plateau where the value of MESC from bending deformations has a maximum. Then it increases again and tends to saturation at tp>>tm.
In manuscript all necessary information is captured by 7 figures. There are 34 references, all of them are adequate and are reflected in the text.
The obtained results are important both for understanding the physical processes that occur in real objects and for the development of new materials. The described manuscript is sufficient, comprehensive and it corresponds to the field of the Journal «Composites Science». It may be accepted after minor revision.